# Analysis of the Spatial and Temporal Changes of NDVI and Its Driving Factors in the Wei and Jing River Basins

**DOI:** 10.3390/ijerph182211863

**Published:** 2021-11-12

**Authors:** Chenlu Huang, Qinke Yang, Weidong Huang

**Affiliations:** 1College of Tourist (Institute of Human Geography), Xi’an International Studies University, Xi’an 710127, China; nwuhcl@163.com; 2College of Urban and Environment Sciences, Northwest University, Xi’an 710127, China; 3Hydrology and Water Resources Bureau of Gansu Province, Lanzhou 730000, China; gsdxhwd@163.com

**Keywords:** Wei River, Jing River, GEE, residual analysis, random forest, classification and regression tree model

## Abstract

This study aimed to explore the long-term vegetation cover change and its driving factors in the typical watershed of the Yellow River Basin. This research was based on the Google Earth Engine (GEE), a remote sensing cloud platform, and used the Landsat surface reflectance datasets and the Pearson correlation method to analyze the vegetation conditions in the areas above Xianyang on the Wei River and above Zhangjiashan on the Jing River. Random forest and decision tree models were used to analyze the effects of various climatic factors (precipitation, temperature, soil moisture, evapotranspiration, and drought index) on NDVI (normalized difference vegetation index). Then, based on the residual analysis method, the effects of human activities on NDVI were explored. The results showed that: (1) From 1987 to 2018, the NDVI of the two watersheds showed an increasing trend; in particular, after 2008, the average increase rate of NDVI in the growing season (April to September) increased from 0.0032/a and 0.003/a in the base period (1987–2008) to 0.0172/a and 0.01/a in the measurement period (2008–2018), for the Wei and Jing basins, respectively. In addition, the NDVI significantly increased from 21.78% and 31.32% in the baseline period (1987–2008) to 83.76% and 92.40% in the measurement period (2008–2018), respectively. (2) The random forest and classification and regression tree model (CART) can assess the contribution and sensitivity of various climate factors to NDVI. Precipitation, soil moisture, and temperature were found to be the three main factors that affect the NDVI of the study area, and their contributions were 37.05%, 26.42%, and 15.72%, respectively. The changes in precipitation and soil moisture in the entire Jing River Basin and the upper and middle reaches of the Wei River above Xianyang caused significant changes in NDVI. Furthermore, changes in precipitation and temperature led to significant changes in NDVI in the lower reaches of the Wei River. (3) The impact of human activities in the Wei and Jing basins on NDVI has gradually changed from negative to positive, which is mainly due to the implementation of soil and water conservation measures. The proportions of areas with positive effects of human activities were 80.88% and 81.95%, of which the proportions of areas with significant positive effects were 11.63% and 7.76%, respectively. These are mainly distributed in the upper reaches of the Wei River and the western and eastern regions of the Jing River. These areas are the key areas where soil and water conservation measures have been implemented in recent years, and the corresponding land use has transformed from cultivated land to forest and grassland. The negative effects accounted for 1.66% and 0.10% of the area, respectively, and were mainly caused by urban expansion and coal mining.

## 1. Introduction

Vegetation is an important “bond” for the interaction of the atmosphere, soil, and water, and it is also an important part of the terrestrial ecosystem [1,2]. Vegetation has the characteristics of intercepting rainfall, slowing surface runoff, and increasing soil infiltration, which play an important role in slowing soil erosion and reducing river sediment content [3]. In the context of global change, vegetation growth can be easily affected by climate change and human activities [4]. The study of vegetation dynamics and the response relationship between climate change and human activities and vegetation has become a key issue [5]. Since 1999, China has implemented a series of soil and water conservation measures such as “returning farmland to forests and grasslands”, and the vegetation cover in the Loess Plateau area has been significantly improved [6,7,8]. At the same time, research related to the spatio-temporal changes of vegetation coverage at global or regional scales has also yielded a series of phased results [9,10,11,12].

Satellite derived vegetation indices (VIs) are broadly used in ecological research, ecosystem services, and land surface monitoring [13,14]. The normalized difference vegetation index (NDVI), perhaps the most utilized VI, implements a remote sensing spectral index for monitoring the land surface. The Landsat-derived NDVI is considered to be one of the best datasets available for long-term NDVI trend analysis because this kind of dataset has a long term (30+ years) and high resolution (30 m), and thus provides a powerful tool to understand vegetation growth history, monitor current conditions, and prepare for future changes. Since 1984, the higher-level surface reflectance products from Landsat sensors, 5 ETM, 7 ETM+, and 8 OLI, have been released [15]. We have used Google Earth Engine (GEE), a planetary-scale cloud-based geospatial analysis platform, which not only can provide users with multi-temporal medium- and high-resolution optical remote sensing data, but also can rapidly process the Landsat data and enable visualization [16]. Using this platform, real-time dynamic monitoring of vegetation conditions in large areas is possible, and is widely used in large-scale vegetation growth monitoring, land use type classification, surface water change analysis, etc. [17,18].

In addition, the relationship between vegetation status and climate change and complex human activities is also currently a popular research topic [19,20,21]. In the past, traditional quantitative methods used the overall information of variables to establish correlation and regression models. However, such methods often ignore the spatial auto-correlation of geographic data [22], and thus cannot accurately describe the true relationship between variables. In particular, in areas where spatial heterogeneity is obvious, this issue is significant [23,24,25]. Machine learning methods such as the Support Vector Machine Framework (SVM), Principal Component Analysis (PCA), Random Forest (RF), and classification and regression tree (CART) provide a novel approach to construct a stable model. Among those approaches, the CART model can reflect the sensitivity of multiple variables to the model fitting results at a spatial scale [26,27].

The Wei and Jing river basins are located in a semi-humid and semi-arid climate zone. In the past, due to their fragile ecological environment and the unreasonable use of land by humans, significant soil erosion occurred in the basin. The intersection of Xianyang Station on the Wei River and Zhangjiashan Station on the Jing River is the “clearly separated area between Jing and Wei River”. Studies have shown that there are significant differences in the annual sediment transport and sediment content of the two river basins, and vegetation cover is a key factor affecting the sediment content [28]. Therefore, this study aimed to analyze the temporal and spatial changes of vegetation cover above the junction of the two watersheds and further explain the “clear and distinct” phenomenon. This study was based on the Landsat surface reflectance product available on the GEE platform. Methods including the Mann–Kendall (MK) trend test and the Pearson correlation coefficient were used to analyze the temporal and spatial changes of NDVI in the Wei River above Xianyang station and Jing River above Zhangjiashan station from 1987 to 2018. Then, based on the random forest and classification and regression tree models, the sensitivity of NDVI to changes in various climate factors was calculated. Finally, the impact of human activities in the two watersheds on NDVI was evaluated through residual analysis.

## 2. Materials and Methods

### 2.1. Study Area Description

The Wei River is the largest tributary of the Yellow River. It originates from Niaoshu Mountain in Weiyuan County, Gansu Province, before flowing through the Longdong Loess Plateau, Tianshui Basin, Baoji Canyon, and Guanzhong Plain. It then flows east to Tongguan County, Weinan City, Shaanxi Province, and merges into the Yellow River. The drainage area is 108,000 km^2^, and the main stream is 818 km long. The upstream flows through the Loess Plateau, so the sediment content is high. The Jing River originates from Jingyuan County and Guyuan County in Liupan Mountain, Ningxia, and flows through the three provinces (regions) of Ningxia, Gansu, and Shaanxi. It then joins the Wei River at Chenjiatan in Gaoling County, Shaanxi Province. The drainage area is 45,421 km^2^, and the main stream is 483 km in length. 

This article focuses on analysis of the areas above Xianyang Hydrological Station on the Wei River and Zhangjiashan Hydrological Station on the Jing River. Xianyang Hydrological Station is the control station for the Jing River to merge into the Wei River. It is 211.1 km from the mouth of the Yellow River and has a catchment area of 48,013 km², accounting for 43.4% of the entire Wei River basin. The amount of sediment is 95.3 million tons, and the annual sediment transport modulus is 2035 t/km^2^. Zhangjiashan Hydrological Station is the Jing River Basin Control Station, 58 km away from the mouth of the Wei River, with a catchment area of 43,789 km^2^, accounting for 40.0% of the entire Wei River basin. It has an annual runoff of 1.566 billion m^3^ and an annual runoff depth of 36.2 mm, with an annual sediment transport volume of 206.4 million tons. The annual sand transport modulus is 4776 t/km^2^. The distribution of river systems in the basin is shown in Figure 1.

### 2.2. Data Sources

(1)Satellite imagery: The Landsat Mission is a long-term (>30 years) high-resolution remote sensing dataset that can provide continuous global historic images. The surface reflectance products have been atmospherically corrected using LEDAPS (Landsat 5, 7) and LaSRC (Landsat 8), and include a cloud, shadow, water, and snow mask produced using CFMASK, in addition to a per-pixel saturation mask. Having a 30 m resolution, this product is ideally suited for local or regional scale time-series applications [13]. All Landsat surface reflectance images (including Landsat 5 ETM, Landsat 7 TM, and Landsat 8 OLI) from 1987 to 2018, with a resolution of 30 m, comprising a total of 5401 images, were retained after removing images with the cloud cover, cloud shadow, water, and snow mask.(2)Climate data: The monthly climate dataset used in this study was the Monthly Climate Water Balance for Global Terrestrial Surface (TerraClimate) dataset [29]. TerraClimate climate data combines the high-resolution (5 km) climate data of WorldClim, and long-term series data of CRU Ts4.0 and Japanese 55 year Reanalysis (JRA55). These data encompass the key elements influencing global land surface energy, including climate variables such as precipitation, temperature, actual evapotranspiration, Palmer drought severity index, and soil moisture in the growing season (April to September) from 1987 to 2018.

### 2.3. Calculation of the NDVI

At present, 40 vegetation indices have been defined [30]. Among these, the normalized difference vegetation index (NDVI) is the most widely used, and is used for research in the fields of global and regional land cover, vegetation classification, and phenological changes. Based on the GEE cloud platform, this study collected Landsat surface reflectance (SR) images, including Landsat 5 ETM, Landsat 7 ETM+, and Landsat 8 OLI/TIRS, and processed the images according to the following rules: (i) The uncertainties of SR in Landsat atmospheric correction and surface reflectance retrieval algorithms are not ideal for water bodies due to the inherently low level of water-leaving radiance, and the consequential very low signal to noise ratio. Similarly, surface reflectance values greater than 1.0 can be encountered over bright targets such as snow and playas. These are known computational artifacts in the Landsat surface reflectance products. Quantitative remote sensing retrievals of water column constituents requires different algorithms, which are being considered for integration into future Landsat surface reflectance products. Landsat 7 ETM+ inputs are not gap-filled in the surface reflectance production. In response to the above problems, users can refer to the Quality Assessment (QA) band for pixel-level condition and validity flags, and select the best available images based on pixel data quality indicators such as cloud or cloud shadow coverage. Based on GEE, we processed images through the function of masksr; this function filters the images by selecting the flags of mask, mask shadow, and water/snow. (ii) Selecting all images in the growing season (April to September) and clipping them according to the study boundary. Then, we resampled all images to a Geographic Coordinate System WGS84 grid of approximately 30 m resolution, and calculated NDVI for selected images based on Formula (1) as the basic data for the research. (iii) The Pearson correlation coefficient method in the GEE platform was used to obtain two important indicators, slope and *p*-value, to assess the trend and significance of the NDVI. The changes in NDVI were divided into the following three types: significant increase (slope > 0, *p*-value < 0.05), insignificant change (slope = 0, slope > 0, slope < 0, *p*-value > 0.05) and significant decrease (slope < 0, *p*-value < 0.05).
NDVI = (NIR − RED)/(NIR + RED)(1)

In the above formula, NIR and RED are near infrared (Landsat 5,7—band 4; Landsat 8—band 5) and infrared (Landsat 5, 7—band 3; Landsat 8—band 4) bands.

### 2.4. Abrupt Change Point Analysis

Goossens (1986) proposed a method to identify the abrupt point of a hydro-meteorological sequence. This method uses sequence reversal to form a hyperbola of the statistic U. When the intersection point is within the confidence interval, it is judged that the corresponding time point is the mutation point [31]. This paper uses the Mann–Kendall mutation point detection method to detect the mutation year when the NDVI changes significantly.

### 2.5. Sensitivity Analysis

Sensitivity analysis (SA) refers to a method that affects the output of the model after some parameters in a mathematical model are changed, and can help researchers to evaluate the influence of parameter estimation on uncertainly and provide a basis for further uncertainty analysis (such as probability analysis). This method was used to analyze the sensitivity of NDVI to various uncertain factors, identify the sensitivity factors and their maximum fluctuation range, and judge the most significant factors affecting NDVI [32]. This study used the following two methods to achieve sensitivity analysis. The first was the random forest method, which was used to assess the importance of variables by establishing a multiple regression model between NDVI and each impact factor. To establish the hierarchical relationships between NDVI and climate factors, the random forest package in the statistical software, R, was used in this study. A random forest is a classifier consisting of tree-structured classifiers, see Formula (2):(2)$$\left\{h(x,\theta_{k}),\;k\;=\;1,\ldots \right\}$$
where the {θ_k_} are independent identically distributed random vectors and each tree casts a unit vote for the most popular class at input x.

Random forests for regression are formed by growing trees depending on a random vector. The output values are numerical and the training set was assumed to be independently drawn from the distribution of the random vector *Y*, *X*. The mean-squared generalization error for any numerical predictor h(x) is:(3)$$E_{X,Y}{\left(Y-h\left(X\right)\right)}^{2}$$

In the evaluation of the random forest model, %IncMSE (Increase in mean squared error (%)), which is the increasing mean squared error rate, is used as the standard to describe the contribution rate of independent variables [33].

The second method involved the construction of a classification and regression tree model (referred to as CART [34]). The algorithm of this model is usually a process of recursively selecting the optimal feature, and segmenting the training data according to the feature, so that each sub-dataset has the best classification process. This process corresponds to the division of the feature space and the construction of the decision tree. CART uses a generalization of the binomial variance called the Gini index. This method first grows an overly large tree and then prunes it to a smaller size to minimize the estimate of misclassification error.
(4)$$\mathrm{Gini}\left(T\right)=1-\sum_{j=1}^{n}p_{j}^{2}$$
where *p* represents the category; *p* refers to the probability of different categories in the dataset sample.

One of the advantages of this method is that the tree node membership can be spatially mapped. In this study, the CART model used to analyze and visualize the interaction between variables was based on R. The variables here relate to the climatic factors, including precipitation, temperature, actual evapotranspiration, Palmer drought severity index and soil moisture, and were extracted based on GEE as auxiliary factors to construct the model. The precipitation is the annual total of the growing season, and the remainder of the climate factors are the annual averages of the growing season. In order to reduce the complexity of the regression tree, the tree depth was set to 3 in this study.

### 2.6. Residual Analysis

In addition to climate factors, human activities are also an important factor affecting vegetation growth. The residual trends method was proposed by Evan and Geerken, and is widely used in related research on how to distinguish the effects of various climate factors and human activity on NDVI at the pixel scale [35]. We used the model parameters derived from a multiple correlation regression as the basis for calculating the predicted value of NDVI, and the difference between the true value of NDVI and the predicted value of NDVI was used as the residual value (Formula (5)):(5)$$\varepsilon \;={\;\mathrm{NDVI}}_{\mathrm{true}}-{\mathrm{NDVI}}_{\mathrm{predicted}}$$

The above formula provides the residual value, ${\;\mathrm{NDVI}}_{\mathrm{true}}$ is the NDVI value calculated based on Landsat surface reflectance data, and ${\mathrm{NDVI}}_{\mathrm{predicted}}$ is the NDVI value calculated after the parameters of the multiple regression model are included. If ε=0, it means that NDVI is affected by climate. If ε>0, it means that human activities play a positive role, otherwise, human activities play a negative role. In GEE, Pearson’s correlation analysis method was used again to establish the correlation between the residual value and the time series, and to estimate the inter-annual change in the residual value. If the residual trend change is not obvious, then the change in NDVI can be explained as the insignificant influence of human activities. On the contrary, the change in NDVI is significantly affected by human activities. The calculation and analysis process of NDVI is shown in Figure 2.

## 3. Results

### 3.1. Variations of NDVI

The analysis of the change trend of the average NDVI value of the Wei and Jing basins from 1987 to 2018 (April–September) shows that the overall NDVI of the Wei and Jing basins showed a significant upward trend from 1987 to 2018. Through the Mann–Kendall trend test method, it was found that the year of NDVI mutation in the basin was 2008 (the intersection of UF and UB) (Figure 3).

Figure 4 shows that the NDVI of the Wei basin is higher than that of the Jing basin overall. The growth rate of the two basins in the base period was basically the same, at 0.0032/a and 0.003/a. After the measurement period, the growth rates of the two basins accelerated to 0.0172/a and 0.01/a, respectively, and the growth rate of the Jing River Basin was greater than that of the Wei River Basin.

### 3.2. Spatial Distribution and Change Characteristics of NDVI

From 1987 to 2018, the spatial distribution of NDVI values in the Wei and Jing basins is shown in Figure 5. The results show that the spatial distribution of NDVI in the two watersheds decreased from southeast to northwest. The NDVI of the Wei River Basin is between 0.0747 and 0.8613, and the NDVI of the Jing River Basin is between 0.1915 and 0.8250. The high values are mainly distributed in the southern part of the study area (the northern foot of the Qinling Mountains), the middle (in the area of Liupan Mountain), the southwestern area near the source of the Weihe River, and the southeastern part of the Jing River Basin, whereas the low values are distributed in the northern and western parts of the study area.

As shown in Figure 6 and Figure 7, the baseline period (1987–2008) is dominated by no significant changes. The significant increase in the areas of the two watersheds was only 21.78% and 31.32%, mainly distributed in the lower reaches of the Jing River and the upper part of the Wei River. The areas showing a significant decrease account for 0.63% and 0.77% in the Wei and Jing basins, respectively, and are mainly distributed in the locations near the urban and coal mining areas. From 1987 to 2018, the NDVI in most areas of the Wei and Jing river basins showed a significant increase; the area percentage of the significant increase was 83.76% and 92.40%, and the area of significant decrease was 0.72% and 0.08%, respectively. The area of the two river basins significantly increased during the measurement period (2008–2018); the NDVI increased by 61.98% and 61.08%, respectively, compared to the baseline period (1987–2008).

### 3.3. The Importance and Sensitivity of Various Climate Factors to NDVI

Table 1 shows the contribution of each covariate in the random forest model, that is, each climatic factor to the NDVI of the model result. The highest contribution is precipitation, having a contribution of 37.05%, followed by soil moisture and temperature, which account for 26.42% and 15.72%, respectively. The contributions of evapotranspiration and the drought index are the smallest, at 12.83% and 9.05% respectively.

Based on the party package in the R language, the climate factors were used as an independent variable to build a classification decision tree, and then the code of each node was extracted and visualized in the form of a raster. Figure 8 and Figure 9 shows the spatial distribution diagram of the decision tree and its nodes established between NDVI and various climate factors. It can be seen from the figure that the two basins are mainly affected by precipitation, among which nodes 4, 5, 7, 8, 11, and 12 are affected by precipitation and soil moisture, and are mainly distributed in the upper reaches of the Wei River and most of the Jing River. The land cover types in these areas are mainly grassland and farmland, and the climate is arid and semi-arid. Thus, the NDVI of these areas is significantly affected by precipitation and soil moisture. Nodes 14 and 15 are affected by precipitation and temperature, and are mainly distributed in the lower reaches of the Wei River. These areas have relatively good hydrothermal conditions, which is conducive to the growth of vegetation.

### 3.4. The Impact of Human Activities and Climate Change on NDVI

The residual trend graph (Figure 10) shows that the impact of human activities on the vegetation cover changes in the Wei and Jing river basins is gradually increasing. In particular, in recent years, human activities have played a significant positive role in the growth of NDVI in the region.

In addition, the spatial distribution map of the residual trend and the statistical table of area proportions (Figure 11, Table 2) show that the proportions of the areas affected by human activities in the two basins reached 80.88% and 81.95%, respectively, of which the proportions of areas with significant positive effects are 11.63% and 7.76%, respectively. These are mainly distributed in the upper reaches of the Wei River, and the western and eastern areas of the Jing River. The significant negative effect areas account for 1.66% and 0.10% respectively, and are distributed in the lower reaches of the Wei River. The corresponding areas are mainly residential areas, and are caused by the expansion of towns.

## 4. Discussion

### 4.1. NDVI in Jing and Wei River

The Wei and Jing river basins are located in the semi-humid to semi-arid climate zone of the Loess Plateau, which is a typical ecologically fragile area in China. This area has lacked vegetation protection for a long time and has serious soil erosion [36]. In the late 1990s, the state accelerated a series of soil and water conservation measures, such as returning farmland to forests/grasslands, which significantly improved the vegetation in most areas of China, and particularly the Loess Plateau [37]. Research by Qiao Chen, Jiang Chong and others found that the middle reaches of the Wei River (mainly the area above Xianyang) has good vegetation coverage. The high values are mainly distributed in the northern foot of the Qinling Mountains, Liupan Mountain, and the southwest corner of the basin near the source area of the Wei River, and the central loess plateau area. The lowest values were previously found in the northern part of the study area [38], which is consistent with the results of this paper. In addition, according to the statistics measured by the hydrological stations, the annual average sediment concentration of the Wei and Jing Rivers are 25 and 132 kg·m^−3^, respectively. The Wei River basin is mostly located in the forest zone, but this is not the case of the Jing River Basin. The research results in this study show that the average NDVI values (1987–2018) of the Wei and Jing Rivers are 0.5463 and 0.4434, respectively, and the measurement of the Wei River forest and grass areas is larger than that of the Jing River basin (Figure 11). Therefore, the difference in vegetation coverage is due to the difference in vegetation coverage in the Wei River basin. The greater amount of sand in the Jing River basin is one of the main reasons for the phenomenon of the “distinction between Jing and Wei”.

Additionally, compared with MODIS NDVI, Landsat NDVI has a longer time series and higher resolution, and can thus more accurately reflect the spatial distribution of surface vegetation and changes over time. This is particularly helpful for the monitoring of vegetation degradation areas (Figure 12 and Figure 13).

### 4.2. The Impact of Climate Factors and Human Activities on Vegetation

Climate change is an important influence on surface vegetation [39,40]. This study found that the precipitation contributes most to NDVI, followed by soil moisture and temperature. The sensitivity analysis showed that small changes in precipitation and soil moisture can cause significant changes in NDVI in the upper and middle reaches of the study area. However, NDVI in the lower reaches of the Wei River is the most sensitive to precipitation and temperature. This is mainly because the middle and upper reaches of the Wei and Jing River basins have an arid climate, low precipitation, and low soil water content, and the water required for plant growth mainly comes from precipitation. Thus, moisture is a key factor that affects the spatial and temporal distribution of vegetation in this area [41]. In addition, the hydrothermal conditions in the study area have gradually improved from northwest to southeast, so the combination of hydrothermal conditions in the lower reaches of the Wei River is better, and changes in temperature and precipitation have a significant impact on vegetation growth.

In addition to climate factors, human activities are also important driving factors affecting vegetation growth [42,43]. The residual analysis results showed that the impact of human activities on vegetation in the Wei and Jing River basins gradually shifted to a positive effect after 2008, and the proportion of the area affected by human activities in the basin reached more than 80%, of which the significant positive effect area accounted for about 9.8%, which is mainly distributed in the upper and middle part of the basin. This is because, since 1999, the Wei and Jing River basins and surrounding areas increased the measures of “returning farmland to forests”, implemented the policy of “closing mountains and grazing prohibition”, and carried out a series of soil and water conservation measures [44,45]. According to the statistics of soil and water conservation measures (Figure 14), the area of forest and grass measures in the Wei and Jing river basins continued to increase from 1954 to 2013. The area of forest and grass in the two basins increased from 141.85 and 126.63 km^2^, to 11,892.32 and 10,904.53 km^2^, from 1954 to 2013. The implementation of forest and grass measures is an important reason for the obvious increase in vegetation cover. In addition, the significant negative effect area in the study area accounts for about 0.90%, and is distributed in the lower reaches of the study area, mainly due to urban expansion and coal mining, which have destroyed the vegetation in the study area.

### 4.3. The Possible Developments and Consequences of This Research

The results of this study show that the average NDVI values (1987–2018) of the Wei and Jing Rivers are 0.5463 and 0.4434, respectively. In addition, according to the statistics measured by the hydrological stations, the annual average sediment concentrations of the Wei and Jing Rivers are 25 and 132 kg·m^−3^, respectively. This indicates the high vegetation coverage and low sand content in the Wei River basin are the main reasons for the phenomenon of “distinction between Jinghe and Weihe”. However, in terms of growth rates, the growth rate of NDVI of Jing River is 0.172/10a, which is 0.1/10a greater than the growth rate of the Wei River basin. In addition, the forest and grass measurements of Jing River also increased rapidly in recent years, and are close to those of the Wei River. Will the phenomenon of “distinction between Jing and Wei River” disappear with the obvious improvement in the vegetation in the Jing River basin? This is a question we need to continue to explore in the future.

Based on the medium- and high-resolution satellite datasets, we analyzed the spatial distribution and changes of the NDVI during the growing season from 1987–2018. The residual analysis and machine learning provided an opportunity to spatially analyze the impact of climate change and human activities on NDVI. This can help us analyze the causes of vegetation degradation in large areas on a more microscopic scale, in order to provide decision support for future vegetation construction planning.

## 5. Conclusions

Based on the Google Earth Engine, machine learning, and other methods, this study analyzed the spatiotemporal changes and driving factors of the NDVI above Xianyang on the Wei River and above Zhangjiashan on the Jing River. The main conclusions are as follows:(1)From 1987 to 2018, the average NDVI values of the Wei and Jing River basins were 0.5463 and 0.4434, respectively. The growth rates of NDVI in the two regions increased from 0.0032/a and 0.003/a in the baseline period (1987–2008) to 0.0172/a and 0.01/a in the measurement period (2008–2018), respectively. Regarding the spatial distribution, the high NDVI values are mainly distributed in the northern foot of the Qinling Mountains, the Liupan Mountains, the southwest corner of the basin near the source of the Wei River, and the Loess Plateau in the middle of the study area. The lowest values are located in the north.(2)Precipitation, soil moisture, and temperature are the three main factors that affect the NDVI in the study area, and the important rates are 37.05%, 26.42%, and 15.72%, respectively. Precipitation and soil moisture are the key factors affecting NDVI in the whole of the Jing River basin and the middle and upper reaches of the Wei River Basin, and precipitation and temperature impact NDVI in the downstream area of Wei River, which has good hydrothermal conditions.(3)After 2008, the impact of human activities on vegetation has gradually become positive. Shares of 80.88% and 81.95% of the Wei and Jing river basins, respectively, have been positively affected by human activities. Among these areas, those with significant positive effects account for 11.63% and 7.76%, respectively, and are distributed in the upper and middle parts of the two basins. The proportions of the areas affected by negative human activities are 1.66% and 0.10%, respectively, and occur in the urban areas and industrial and mining land.

## Figures and Tables

**Figure 1 ijerph-18-11863-f001:**
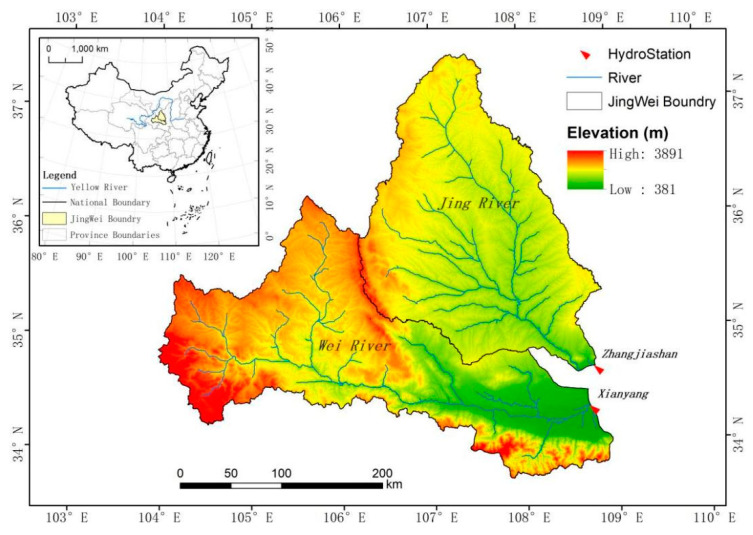
Location of the study area and distribution of the hydrostations in the Wei and Jing River Basin.

**Figure 2 ijerph-18-11863-f002:**
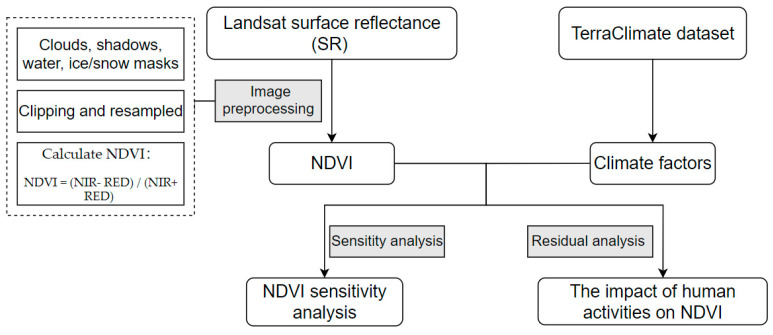
A flow chart of NDVI calculation, and the correlation analysis of human activities and climate change on NDVI.

**Figure 3 ijerph-18-11863-f003:**
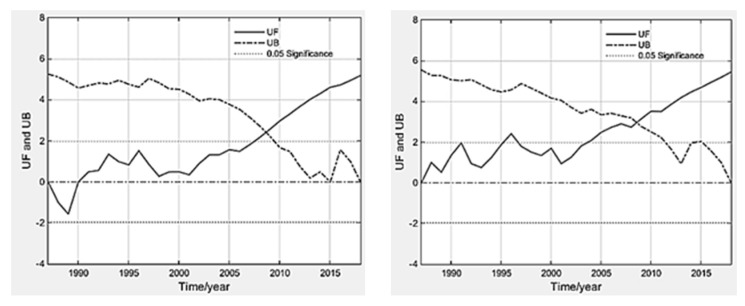
Abrupt point of NDVI in Wei and Jing River Basin during 1987–2018 (left-hand side shows abrupt detection in Wei River Basin, right-hand side shows that in Jing River Basin).

**Figure 4 ijerph-18-11863-f004:**
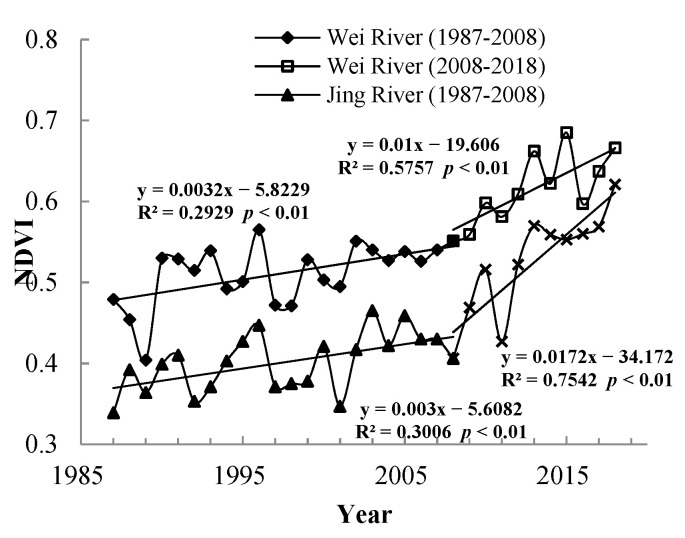
Trend of NDVI in Wei and Jing River Basin during 1987–2018.

**Figure 5 ijerph-18-11863-f005:**
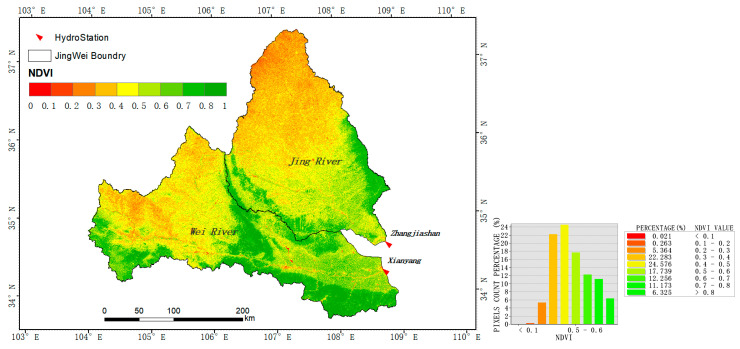
Spatial distribution of NDVI (mean value in growth season) from 1987 to 2018 in Wei and Jing River Basin.

**Figure 6 ijerph-18-11863-f006:**
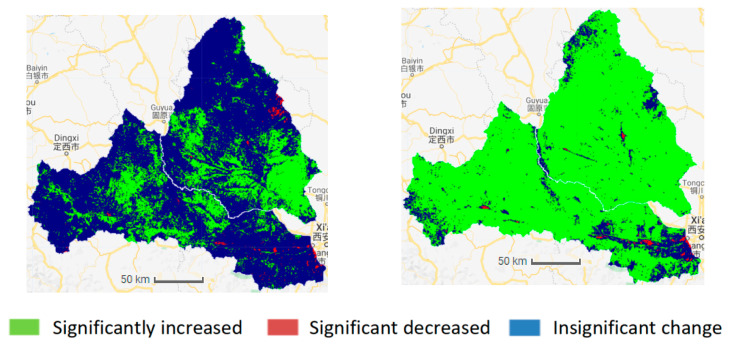
The significant change in NDVI in the Wei and Jing River Basin (**Left**: 1987–2008, **Right**: 1987–2020).

**Figure 7 ijerph-18-11863-f007:**
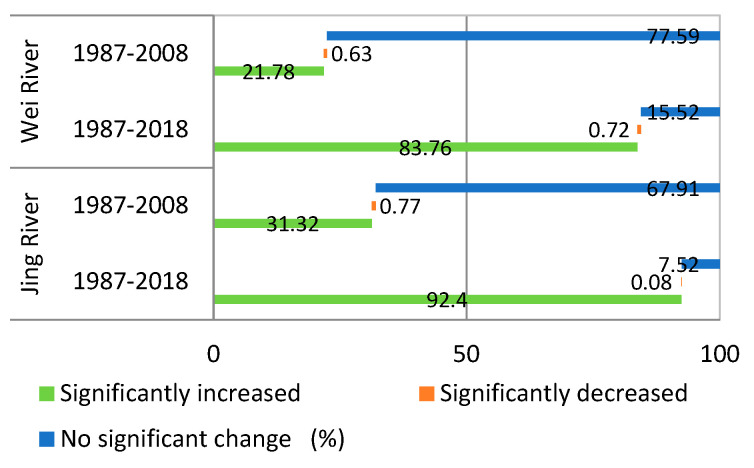
The proportion of NDVI significant change area in the Wei and Jing River Basin.

**Figure 8 ijerph-18-11863-f008:**
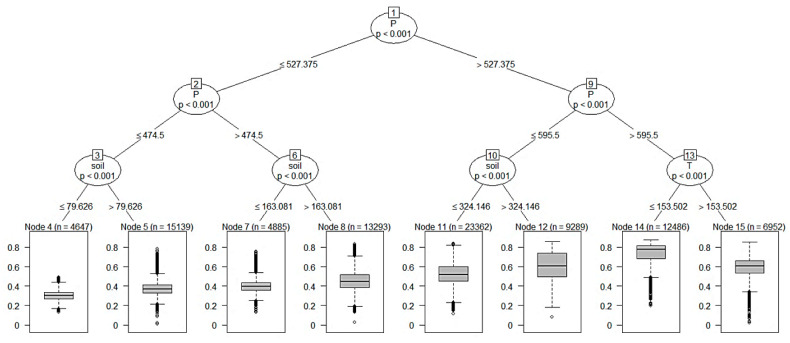
NDVI factor interactions derived from the classification and regression tree (CART) model. Note: In the above figure, the dataset is represented as a tree plot (upper part), reporting the importance of the interactions between factors with a downwards trend. The factors are as follows: *p* is the total precipitation during the growing season, soil is the average soil moisture during the growing season, and T is the average temperature during the growing season. Additionally, the boxplot at the terminal nodes level (lower part) shows how their interaction affects the estimated NDVI.

**Figure 9 ijerph-18-11863-f009:**
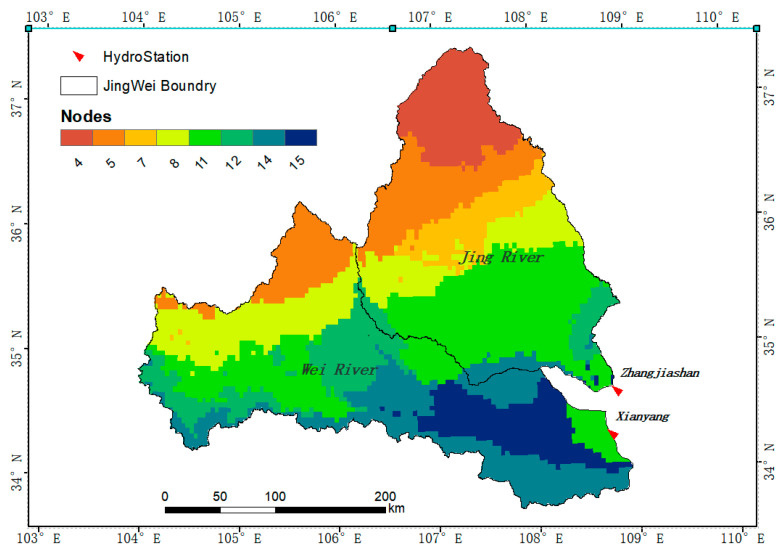
Spatial representation of the nodes’ membership derived from the sensitively analysis.

**Figure 10 ijerph-18-11863-f010:**
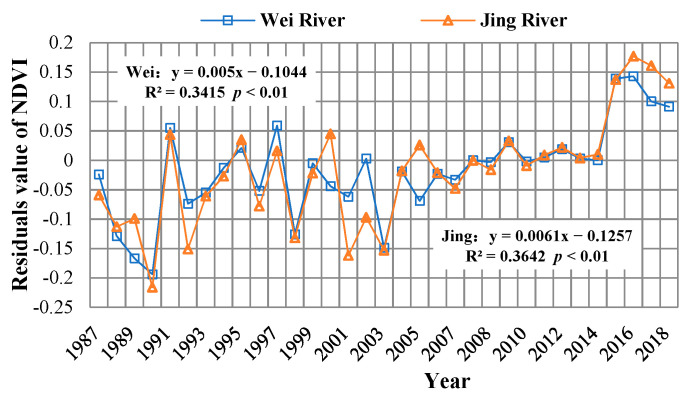
The trend of NDVI residuals in Wei and Jing River basins during 1987–2018.

**Figure 11 ijerph-18-11863-f011:**
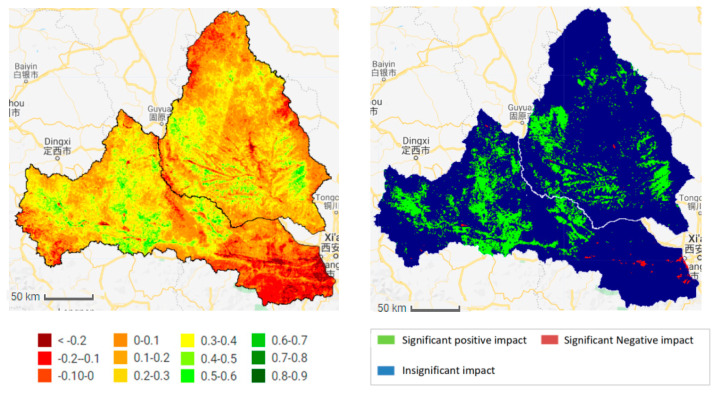
The spatial distribution of NDVI residuals in Wei and Jing River basins during 1987–2018 (**Left**: Trend, **Right**: Significant change).

**Figure 12 ijerph-18-11863-f012:**
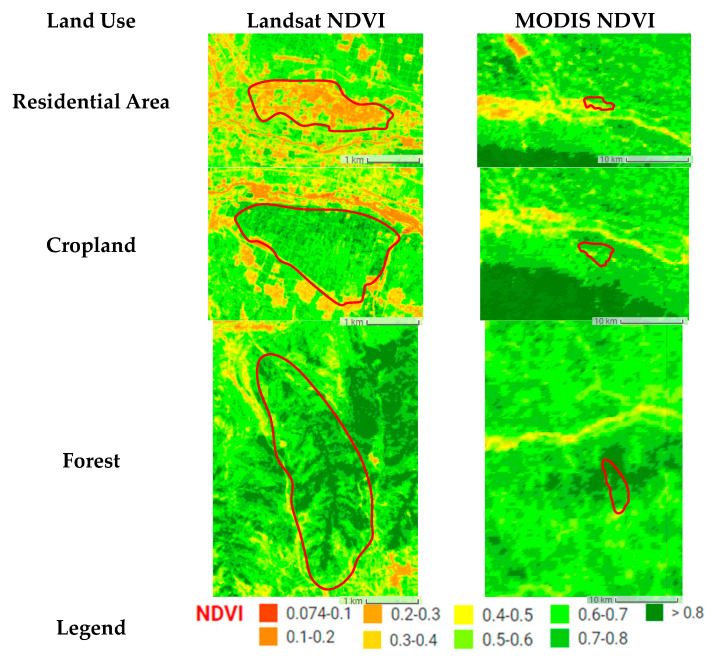
The comparison diagram of spatial distribution of Landsat NDVI and MODIS NDVI (2010).

**Figure 13 ijerph-18-11863-f013:**
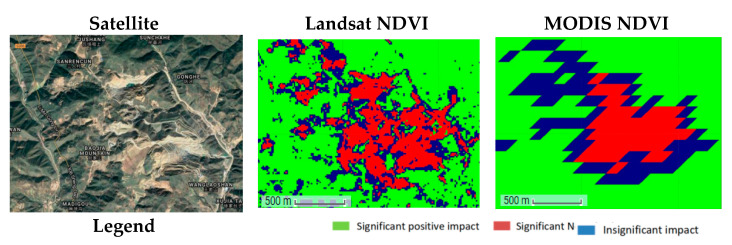
The comparison diagram of significant change in Landsat NDVI and MODIS NDVI (2000–2018).

**Figure 14 ijerph-18-11863-f014:**
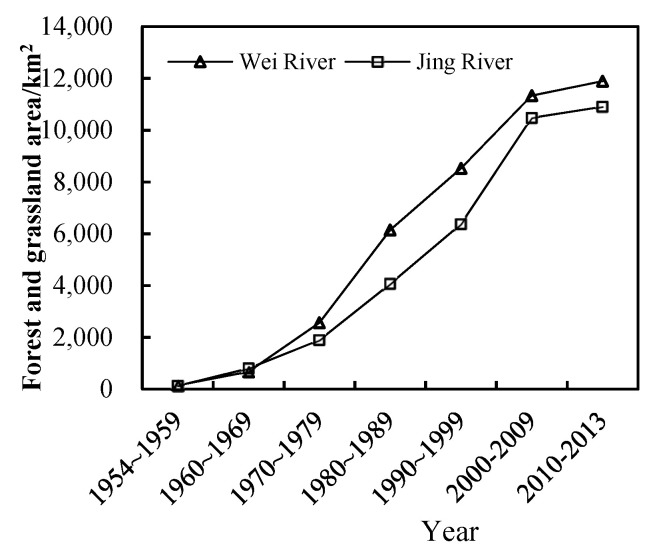
The area of forest and grassland implemented in Wei and Jing River basins during 1954–2013.

**Table 1 ijerph-18-11863-t001:** The variance importance of climate factors.

Covariate	Precipitation (P)	Soil Moisture (Soil)	Temperature (T)	Evapotranspiration (AET)	Drought Index (pdsi)
%IncMSE	37.05	26.42	15.72	12.83	9.05

**Table 2 ijerph-18-11863-t002:** The residual area and proportion of Wei and Jing basins from 1987 to 2018.

	ε > 0		ε < 0	
	Area/km^2^	Proportion/%	Area/km^2^	Proportion/%
Wei	39,345.34	81.95	8667.7	18.05
Jing	35,416.49	80.88	8372.44	19.12
	**Significant Position Impact**		**Significant Negative Impact**	
	**Area/km^2^**	**Proportion/%**	**Area/km^2^**	**Proportion/%**
Wei	4575.64	9.53	144.04	0.3
Jing	2749.94	6.28	8.76	0.02

Note: ε is the residual value.

## Data Availability

All data, models, and code that support the findings of this study are available from the corresponding author upon reasonable request.

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
