# Peer review of "Analysis of the Spatial and Temporal Changes of NDVI and Its Driving Factors in the Wei and Jing River Basins"

_ijerph, 2021, doi:10.3390/ijerph182211863_

Round 1
Reviewer 1 Report
This is a well-written paper. Unfortunately, it covers the topic that has been published so much time that at this point I question the point of this paper. Yes this is an well made case study but offers nothing new to the method and only confirms known facts. I will live it to the editor if he wants this paper published or not.
Some minor comments
L16 chenge ‘established’ to ‘used’
L17 ‘Then, further study 17 the impact of human activities on NDVI based on the residual analysis method.’ – fic grammar in this sentence
L22 remove incised
- Number of images?
L179 – I would suggest using europenian letters true the text unless the annotations are explained
L303 “Climate change affects the change of surface vegetation”
Author Response
- L16 change ‘established’ to ‘used’
Re: We thank the reviewer for this comment. We revised as suggested in the line 16.
- L17 ‘Then, further study 17 the impact of human activities on NDVI based on the residual analysis method.’ – fic grammar in this sentence
Re: We revised this sentence as “Then, based on the residual analysis method, the effects of human activities on NDVI were explored”. See line 18.
- L22 remove incised
Re: Done as suggested.
- Number of images?
Re: The images in this article are 11.
- L179 – I would suggest using europenian letters true the text unless the annotations are explained
Re: Thanks for your suggestion, we revised the equation in manuscript.
- L303 “Climate change affects the change of surface vegetation”
Re: L303 here want to illustrate the Landsat NDVI can reflect the spatial distribution of surface vegetation through compared with MODIS NDVI. And I guess the reviewer have some questions in Line 309, because the Line 309 show “Climate change is an important reason affecting the change of surface vegetation”, we change this sentence as “Climate change is an important reason affecting the surface vegetation” by remove “change of”. See line 333.

Reviewer 2 Report
(1) The methods of this manuscript is poorly presented, and many of them are unclear, for example, the calculation of NDVI based on satellite derived datasets. Further, how the satellite datasets are utilized are poorly described. The authors should mention their procedures in step-by-step manner.
(2) Section 2.4: Sensitivity Analysis only mentioned a lot of different mathematical / statistical methods, however didn't go into deep. Readers may easily get confused because of this.
(3) Is the implication of residual analysis based on previous literature, or simply what the authors concluded / preset? If so, please justify it!
(4) Figure 6 has to be better presented.
(5) Resolutions of Figures 7a and b are too low. need to be more clear for readers. Also, it's better to give a summary of the box and whisker diagram figures of Figures 7a.
(6) Figure 8: need to be more professional, in terms of removing grid, font style etc.
(7) The conclusion can be shortened, currently, it is too long.
(8) Methodology of this manuscript is quite problematic, as the authors didn't provide any fine details of data handling, projection, NDVI calculations etc. We hope the readers can link up all these in a coherent manner, otherwise the paper cannot be centralized.
Author Response
1 The methods of this manuscript is poorly presented, and many of them are unclear, for example, the calculation of NDVI based on satellite derived datasets. Further, how the satellite datasets are utilized are poorly described. The authors should mention their procedures in step-by-step manner.
Re:We added and adjusted the details about NDVI descriptions and calculation in section of 2.3 as following(the text with gray background is modified):
At present, 40 vegetation indices have been defined[30]. Among them, the normalized difference vegetation index (NDVI) is the most widely used, which is used for research in the fields of global and regional land cover, vegetation classification, and phenological changes. Based on the GEE cloud platform, this paper collected Landsat surface reflectance (SR) images including Landsat 5 ETM, Landsat 7 ETM+ and Landsat 8 OLI/TIRS, and process the images according to the following rules: (i) Selecting best available images based on pixel data quality indicators such as cloud or cloud shadow coverage, water, ice and snow cover; (ii) Selecting all images in the growing season (April to September) and clipping them according to the study boundary. Then, resampled all images to a Geographic Coordinate System WGS84 grid of approximately 30m resolution, and calculated NDVI for selected images based on formula (1) as the basic data for the research; (iii) The Pearson correlation coefficient method in the GEE platform is used to obtain two important indicators, slope and p-value, to assess the trend and significance of the NDVI. The changes in NDVI are divided into the following three types: significant increase (slope> 0, p-value <0.05), insignificant change (slope = 0, slope> 0, slope <0, p-value> 0.05) and significant decrease (slope <0, p -value <0.05).
NDVI = (NIR- RED) / (NIR+ RED) (1)
- Section 2.4: Sensitivity Analysis only mentioned a lot of different mathematical / statistical methods, however didn't go into deep. Readers may easily get confused because of this.
Re: We added more reference and description for sensitivity analysis method in 2.4 section as following (the text with gray background is modified):
Sensitivity analysis (SA) refers to a method that affects the output of the model after some parameters in a mathematical model are changed[32]. This study uses the following two methods to achieve sensitivity analysis. The first is the random forest method, which is used to assess the importance of variables by establishing a multiple regression model between NDVI and impact factors. In the evaluation of the random forest model, %IncMSE (Increased in mean squared error(%)), which is the increasing mean squared error rate, is used as the standard to describe the contribution rate of independent variables[33]. The second method is to construct a classification and regression tree model (Classification and Regression Tree, referred to as CART [34]). The algorithm of this model is usually a process of recursively selecting the optimal feature, and segmenting the training data according to the feature, so that each sub-data set has the best classification process. This process corresponds to the division of the feature space and the construction of the decision tree. One of the advantages of this method is that the tree node membership can be spatially mapped. In this study, the authors used CART model to analyze and visualize the interaction between variables. The variables here choose the climatic factors including Precipitation, Temperature, Actual evapotranspiration, Palmer Drought Severity Index) and Soil moisture were extracted based on GEE as auxiliary factors to construct the model. The precipitation is the annual total of the growing season, and the rest of the climate factors are the annual averages of the growing season. In order to reduce the complexity of the regression tree, the tree depth was set as 3 in this study.
- Is the implication of residual analysis based on previous literature, or simply what the authors concluded / preset? If so, please justify it!
Re: The residual analysis refers to the previous literature. This methods was proposed by Evan and Geerken, and widely used in related research on how to distinguish the effects of various climate factors and human activity on NDVI at pixel scale. For example, Jiang[1] explore the vegetation dynamics and responses to climate change and human activities in Central Asia; Qu[2] answer a question about “What drives the vegetation restoration in Yangtze River basin, China: Climate change or human activities?” by using residuals analysis.
- Jiang, C.; Xiong, L.; Wang, D.; Liu, P.; Guo, S.; Xu, C.Y. Separating the impacts of climate change and human activities on runoff using the Budyko-type equations with time-varying parameters. Journal of Hydrology 2015, 522, 326–338.
- Qu, S.; Wang, L.; Lin, A.; Zhu, H.; Yuan, M. What drives the vegetation restoration in Yangtze River basin, China: Climate change or anthropogenic factors? Ecological Indicators 2018, 90, 438–450.
- Figure 6 has to be better presented.
Re: Thanks for your suggestion, in order to make the picture better, we change the Figure.6 in the manuscipt.
Compared with the former one, the revised figure can more clearly and intuitively show the label of significant changes in the NDVI of two watersheds.
- need to be more clear for readers. Also, it's better to give a summary of the box and whisker diagram figures of Figures 7a.
Re: Thanks for your notice. We add some descriptions including the meaning of box and whisker diagram figures below the Figure7a.
- Figure 8: need to be more professional, in terms of removing grid, font style etc.
Re: We have removed the grid and the labels of latitude and longitude of Figure 8.
- The conclusion can be shortened, currently, it is too long.
Re: Thank you so much for your suggestion. We further revised and simplified the conclusion in the lines 372-392.
- Methodology of this manuscript is quite problematic, as the authors didn't provide any fine details of data handling, projection, NDVI calculations etc. We hope the readers can link up all these in a coherent manner, otherwise the paper cannot be centralized.
Re: We thank the reviewer for this comment. In order to make the methods more detailed and clear, we add the information including the projection and resolution of images and explain the calculation process of NDVI in the form of a chart, the flowchart.

Reviewer 3 Report
The paper is interesting. The research proposed by the authors is finalized to analyze the temporal and spatial changes of vegetation cover above the junction of the two water- sheds. This study is based on the Landsat surface reflectance product available on the GEE platform, some methods are used as Mann-Kendall (MK) trend test method and Pearson correlation coefficient, to analyze the temporal and spatial changes of NDVI (in the Wei River and Jing River from 1987 to 2018); based on random forest, classification and regression tree models, the sensitivity of NDVI to changes in various climate factors is calculated, and the impact of human activities in the two water-sheds on NDVI is evaluated through residual analysis.
Some additions are required in order to improve the clarity of this contribution.
- The pedic in formula 2 should be in English.
- The caption in Figure 2 is not clear
- Authors should improve subsection 3.3 and the description of nodes.The text in lines 242-247 is not clear, it should be deepened the characterization of the various nodes, otherwise it is not clear how you get to Figure 7.
- What are the possible developments of this research.
- What will be the consequences of this study.
- Supplement the bibliographical references.
Author Response
- The pedic in formula 2 should be in English.
Re: Done as suggested. See line 196.
- The caption in Figure 2 is not clear
Re: Thanks for your comments, we recreated a cleared picture in the new version manuscript.
- Authors should improve subsection 3.3 and the description of nodes.The text in lines 242-247 is not clear, it should be deepened the characterization of the various nodes, otherwise it is not clear how you get to Figure 7.
Re: Thanks for your suggestion. Regarding the description of the node, we have added a more detailed description, including the distribution position and the meaning of the representative of the node. The revised content is as follows (the text with gray background is modified)):
“Figure 7 shows the spatial distribution diagram of the decision tree and its nodes established between NDVI and various climate factors. It can be seen from the figure that the two basins are mainly affected by precipitation, among which nodes 4, 5, 7, 8, 11, and 12 are affected by precipitation and soil moisture, and are mainly distributed in the upper reaches of the Wei River and most of the Jing River. The Land cover types in these areas are mainly grassland and farmland, and the climate is arid and semi-arid. So, the NDVI of these areas is significantly affected by precipitation and soil moisture. Nodes 14 and 15 are affected by precipitation and temperature and are mainly distributed in the lower reaches of the Wei River. These areas have relatively good hydrothermal conditions, which is conducive to the growth of vegetation.”
Regarding the process of obtaining nodes, we have added the detail process of obtaining nodes as following: “Based on the party package in the R language, the climate factors are used as an independent variable to build a classification decision tree, and then the code of each node is extracted and them visualized in the form of a raster.” We added this part to the section 3.3. See lines 262-264.
- What are the possible developments of this research.
Re: The results of this paper show that the average NDVI values (1987-2018) of the Wei River and Jing River are 0.5463 and 0.4434 respectively. Besides, according to the statistics measured by the hydrological station, the annual average sediment concentration of the Wei River and Jing River are 25 kg·m-3 and 132 kg·m-3, respectively. This explains high vegetation coverage and low sand content in the Wei River Basin are the main reasons for the phenomenon of "distinguishment between Jinghe and Weihe". However, in terms of growth rate, the growth rate of NDVI of Jing River is 0.172/10a, which is greater than the growth rate of Wei River Basin by 0.1/10a, and the forest and grass measures of Jing River has also increased rapidly in recent years, even close to that of the Wei River. Will the phenomenon of "distinguishment between Jing and Wei River" disappear with the obvious improvement of the vegetation in the Jing River Basin? "This is a question we need to continue to explore in the future.
- What will be the consequences of this study.
Re: Based on the medium-high resolution satellite datasets, we analysis the spatial distribution and changes of growing season NDVI from 1987-2018. The residual analysis and machine learning give us an opportunity to spatially analysis the impact of climate change and human activities on NDVI. This can help us analyze the causes of vegetation degradation in large areas on a more microscopic scale, in order to provide decision support for future vegetation construction planning.
- Supplement the bibliographical references.
Re: Thanks for your suggestion. We have added references in the manuscript in the discussion section.

Round 2
Reviewer 2 Report
The authors have improved their writing style, and have provided more information regarding the methodology and scheme of work. However, there are several more major issues to consider before further proceeding this manuscript:
(1) The authors have only adopted random forest and classification, and regression techniques to retrieve NDVI or vegetation indices from Landsat satellite derived datasets. However, one recent work has shown that Support Vector Machine (SVM) framework could show very good results in retrieving land use features and spatial distribution of vegetation land within developing city. Please refer to: https://doi.org/10.3390/rs13163337
We think the authors should compare the statistical performance (in particular the maps retrieved in Figure 1 - spatial distribution of land use patterns) of different algorithms, in particular, SVM in the above paper, and see if CART (the one mentioned in this manuscript) is good enough for conducting larger spatial retrieval process. The authors should also explain the pros and cons of using both SVM (in the link above) and CART (in the manuscript) for similar retrieval processes.
(2) For sensitivity analysis (Section 3.3), the authors should clearly state the statistical and mathematical formula adopted, before showing numerical figures like Table 1.
(3) The numerical figures in Figure 4 and Figure 9 (the slopes, y-intercepts etc. of the best fit line) should be correct to 3 or 4 sig. fig., rather than having inconsistent numerical formats.
(4) In Section 2.2, the authors simply state the data sources used in this study. However, are these datasets the best (or the most clear datasets)? The authors should mention about the reliability and application of these datasets, before using them for conducting NDVI retrieval and calculation.
(5) Font size of Figure 1, Figure 5, Figure 8b) should be consistent, currently there are many font styles in it.
(6) Figure 2: What kind of image processing techniques have the authors used? Please state it more clearly.
Therefore, I suggest the authors to make all aforementioned changes before re-submitting the manuscript for consideration.
Author Response
(1) The authors have only adopted random forest and classification, and regression techniques to retrieve NDVI or vegetation indices from Landsat satellite derived datasets. However, one recent work has shown that Support Vector Machine (SVM) framework could show very good results in retrieving land use features and spatial distribution of vegetation land within developing city. Please refer to: https://doi.org/10.3390/rs13163337
We think the authors should compare the statistical performance (in particular the maps retrieved in Figure 1 - spatial distribution of land use patterns) of different algorithms, in particular, SVM in the above paper, and see if CART (the one mentioned in this manuscript) is good enough for conducting larger spatial retrieval process. The authors should also explain the pros and cons of using both SVM (in the link above) and CART (in the manuscript) for similar retrieval processes.
Re: We thank the reviewer for this comment. According to reviewer’s suggestion, we read article carefully, and we found that the Support Vector Machine (SVM) method was originally developed for binary classification, which is applicable for the classification of land-use types, as well as detecting temporal and spatial changes of land covers. For our study, we need analyze and visualize the interaction between variables and NDVI. The SVM can help us realize the binary classification, but it lacks the function of displaying the tree node spatially. So, we think the SVM may not be suitable for this research. However, this is indeed a good research idea. In future research, we will focus on the similarities and differences between SVM and CART.
- For sensitivity analysis (Section 3.3), the authors should clearly state the statistical and mathematical formula adopted, before showing numerical figures like Table 1.
Re: In the section of sensitivity analysis, some theoretical background and mathematical formulas for random forest and CART were added. See lines 190-202, 211-216.
- The numerical figures in Figure 4 and Figure 9 (the slopes, y-intercepts etc. of the best fit line) should be correct to 3 or 4 sig. fig., rather than having inconsistent numerical formats.
Re: The numerical formats of slopes, y-intercepts, x-axis, y-axis have been unified in Figure 4, Figure 9, Figure 12.
- In Section 2.2, the authors simply state the data sources used in this study. However, are these datasets the best (or the most clear datasets)? The authors should mention about the reliability and application of these datasets, before using them for conducting NDVI retrieval and calculation.
Re: According to the advantage of the datasets, we add some information in the manuscript as following: “The Landsat Mission is the long-term (>30 years) high-resolution remote sensing dataset that can provide a continuous historic images globally. The surface reflectance products have been atmospherically corrected using LEDAPS (Landsat 5,7 ) and LaSRC (Landsat 8), and include a cloud, shadow, water and snow mask produced using CFMASK, as well as a per-pixel saturation mask. This product at 30-m resolution is ideally suited for local or regional scale time-series applications[13].”, see lines 128-134.
- Font size of Figure 1, Figure 5, Figure 8b) should be consistent, currently there are many font styles in it.
Re: Thanks for your suggestion. We set the font size, scale bar in the Figure 1, Figure 5 and Figure 8 to be consistent in this manuscript. In order to make Figure 5 more clear, we put the table in the original figure alone on the right side of the figure 5.
- Figure 2: What kind ofimage processing techniques have the authors used? Please state it more clearly.
Re: For image processing, we mainly use the function of masksr, which mask clouds/cloud shadow based on the pixel_qa band of Landsat SR data, the code as following:
function masksr(image) {
// Bits 3 and 5 are cloud shadow and cloud, respectively.
var cloudShadowBitMask = (1 << 3);
var cloudsBitMask = (1 << 5);
// Get the pixel QA band.
var qa = image.select('pixel_qa');
// Both flags should be set to zero, indicating clear conditions.
var mask = qa.bitwiseAnd(cloudShadowBitMask).eq(0)
.and(qa.bitwiseAnd(cloudsBitMask).eq(0));
return image.updateMask(mask);
}
Therefore, I suggest the authors to make all aforementioned changes before re-submitting the manuscript for consideration.

Reviewer 3 Report
With reference to the following points, I cannot see the integration in the text. Maybe the integrations are in the notes that I can’t see in the pdf file?
With reference to the first question of the next point now figure 7 is fine.
From the authors cover letters:
“3. Authors should improve subsection 3.3 and the description of nodes.The text in lines 242-
247 is not clear, it should be deepened the characterization of the various nodes, otherwise it
is not clear how you get to Figure 7.
Re: Thanks for your suggestion. Regarding the description of the node, we have
added a more detailed description, including the distribution position and the
meaning of the representative of the node. The revised content is as follows (the text
with gray background is modified)):
“Figure 7 shows the spatial distribution diagram of the decision tree and its
nodes established between NDVI and various climate factors. It can be seen from the
figure that the two basins are mainly affected by precipitation, among which nodes 4,
5, 7, 8, 11, and 12 are affected by precipitation and soil moisture, and are mainly
distributed in the upper reaches of the Wei River and most of the Jing River. The
Land cover types in these areas are mainly grassland and farmland, and the climate
is arid and semi-arid. So, the NDVI of these areas is significantly affected by
precipitation and soil moisture. Nodes 14 and 15 are affected by precipitation and
temperature and are mainly distributed in the lower reaches of the Wei River. These
areas have relatively good hydrothermal conditions, which is conducive to the
growth of vegetation.”
Regarding the process of obtaining nodes, we have added the detail process
of obtaining nodes as following: “Based on the party package in the R language, the
climate factors are used as an independent variable to build a classification decision
tree, and then the code of each node is extracted and them visualized in the form of a
raster.” We added this part to the section 3.3. See lines 262-264.
- What are the possible developments of this research.
Re: The results of this paper show that the average NDVI values (1987-2018) of the
Wei River and Jing River are 0.5463 and 0.4434 respectively. Besides, according to
the statistics measured by the hydrological station, the annual average sediment
concentration of the Wei River and Jing River are 25 kg·m-3 and 132 kg·m-3,
respectively. This explains high vegetation coverage and low sand content in the Wei
River Basin are the main reasons for the phenomenon of "distinguishment between
Jinghe and Weihe". However, in terms of growth rate, the growth rate of NDVI of
Jing River is 0.172/10a, which is greater than the growth rate of Wei River Basin by
0.1/10a, and the forest and grass measures of Jing River has also increased rapidly in
recent years, even close to that of the Wei River. Will the phenomenon of
"distinguishment between Jing and Wei River" disappear with the obvious
improvement of the vegetation in the Jing River Basin? "This is a question we need
to continue to explore in the future.
- What will be the consequences of this study.
Re: Based on the medium-high resolution satellite datasets, we analysis the spatial
distribution and changes of growing season NDVI from 1987-2018. The residual
analysis and machine learning give us an opportunity to spatially analysis the
impact of climate change and human activities on NDVI. This can help us analyze
the causes of vegetation degradation in large areas on a more microscopic scale, in order to provide decision support for future vegetation construction planning”.
With regard to all other integration, the authors have improved the level of the paper.
As soon as you have checked the questions of additions concerning the points highlighted, the paper can be considered accepted for publication.
Author Response
With reference to the following points, I cannot see the integration in the text. Maybe the integrations are in the notes that I can’t see in the pdf file?
Re: We thank the reviewer for this comment. For question 3, we added the information in the manuscript in the lines 299-301, 306-311.
For the remaining two questions, we have explained in cover letter without adding to the manuscript. We added the integration in section 4.3 (Lines 405-423). Thank you so much for your reminder.

Round 3
Reviewer 2 Report
Thanks for revising the manuscript. There are several points that the authors should further take note of, and revise accordingly:
(1) For point 1: The use of SVM can help us realizing binary classification, and there are two ways of incorporating SVM into this manuscript: (a) The authors should acknowledge the potential usage of SVM-based algorithm for land retrieval, especially for spatial distribution of vegetation land, as mentioned in "https://doi.org/10.3390/rs13163337", within appropriate sections of the manuscript; (b) The authors should discuss the similarities and differences between SVM (and/or other machine learning methodologies / statistical algorithms) and CART (the way they proposed in this paper). Currently, the authors simply focus on developing CART in this paper, but without any proper acknowledgment of other similar studies / methods. The authors should add it, and the potential combination of different statistical techniques in the revised manuscript.
(2) For reliability of datasets used in this study: The authors should not only mention the history of Landsat mission, correction of surface reflectance datasets, and adjustments imposed, they should also cite relevant references or previous papers that discuss the sensitivity or potential errors of such datasets. Also, they should explain how input datasets of these satellite missions affect the eventual satellite-derived products, and the environmental factors that would affect the accuracy of retrieval within this study. More discussion should be added, and the authors should add more details into the section of "sensitivity analysis" of current study as well.
(3) The image processing techniques (Lines 150-162 of the current version): The authors should not include any codes in their manuscript. Instead, they should convert the techniques into words or mathematical equations, which could be easily understood by layman. The authors should also explain the purpose of each step of their codes (Lines 150-162).
(4) Figure 3 (Lines 263-266) is not professional, the authors should use consistent graphical software to obtain these types of graphs. Better follow the format of Figure 4. Further, the font size and style, labeling of y-axis of Figure 4 is inconsistent and problematic. Consistent font size, font style and positions of axes labeling should be used within the entire manuscript.
(5) Figures 8(a) and 8(b) should be separated into 2 figures, because one is more on algorithmic explanation, and another on spatial plot.
(6) Lines 369-372: Additionally, compared with MODIS NDVI, Landsat NDVI has a longer time series and higher resolution, which can more accurately reflect the spatial distribution of surface vegetation and changes over the years, especially helpful for the monitoring of vegetation degradation areas - could the authors add in the exact spatial locations and geographical areas of these "vegetation degradation areas"?
(7) Lines 422-424: The residual analysis and machine learning give us an opportunity to spatially analysis the impact of climate change and human activities on NDVI - How could this be conducted? It seems that the authors have not mentioned the potential extension in an explicit manner. They just outline the idea, but without providing much details. That's not a good practice for writing academic paper.
(8) Line 445: How do you define "important rate"? Any formula in the main text / Methodology?
(9) The motivation of this study is not well explained in the "Introduction". Also, the Introduction should be re-written so that the intuition, motivation and ideas of development of this research could be more obviously illustrated. Further, is there any similar study conducted before? If so, please include them in the Introduction. Also, the structure of this manuscript should also be included in the last paragraph of Section I. Introduction. Please reformat the entire Introduction section.
(10) Section 4.3: The "consequences" of this research are not clearly explained. Do you actually mean "benefits" of this research in interdisciplinary areas of vegetation, land use distribution, etc.?
Finally, the authors should proceed for grammatical checking before re-submission.
Author Response
Thanks for revising the manuscript. There are several points that the authors should further take note of, and revise accordingly:
(1) For point 1: The use of SVM can help us realizing binary classification, and there are two ways of incorporating SVM into this manuscript: (a) The authors should acknowledge the potential usage of SVM-based algorithm for land retrieval, especially for spatial distribution of vegetation land, as mentioned in "https://doi.org/10.3390/rs13163337", within appropriate sections of the manuscript; (b) The authors should discuss the similarities and differences between SVM (and/or other machine learning methodologies / statistical algorithms) and CART (the way they proposed in this paper). Currently, the authors simply focus on developing CART in this paper, but without any proper acknowledgment of other similar studies / methods. The authors should add it, and the potential combination of different statistical techniques in the revised manuscript.
Re: Thanks for your comments. Your suggestion give us a good way to solute this question, because it’s little hard to add new method in this manuscript. So, we have added some statistical method that used in other researches, and discussion the similarities and differences between SVM (and/or other machine learning methodologies / statistical algorithms) with CART model. The content as following: “Machine learning methods such as Support Vector Machine Framework (SVM), Principal Component Analysis (PCA), Random Forest(CF) and Classification and Regression Tree(CART) provide a novel approach way to construct a stable model, among those approaches, CART model can reflect the sensitivity of multiple variables to the model fitting results on a spatial scale. “
(2) For reliability of datasets used in this study: The authors should not only mention the history of Landsat mission, correction of surface reflectance datasets, and adjustments imposed, they should also cite relevant references or previous papers that discuss the sensitivity or potential errors of such datasets. Also, they should explain how input datasets of these satellite missions affect the eventual satellite-derived products, and the environmental factors that would affect the accuracy of retrieval within this study. More discussion should be added, and the authors should add more details into the section of "sensitivity analysis" of current study as well.
Re: Thanks for your comments. Landsat Surface reflectance (SR) improves comparison between multiple images over the same region by accounting for atmospheric effects such as aerosol scattering and thin clouds, which can help in the detection and characterization of Earth surface change. Surface reflectance is the amount of light reflected by the surface of the Earth. It is a ratio of surface radiance to surface irradiance, and as such is unitless, with values between 0 and 1. The uncertain of SR is Landsat atmospheric correction and surface reflectance retrieval algorithms are not ideal for water bodies due to the inherently low level of water leaving radiance, and the consequential very low signal to noise ratio. Similarly, surface reflectance values greater than 1.0 can be encountered over bright targets such as snow and playas. These are known computational artifacts in the Landsat surface reflectance products. Quantitative remote sensing retrievals of water column constituents requires different algorithms, which are being considered for integration into future Landsat surface reflectance products. In response to the above problems, users can refer to the Quality Assessment(QA) band for pixel-level condition and validity flags.and select best available images based on pixel data quality indicators such as cloud or cloud shadow coverage. See lines 153-166.
Sensitivity analysis (SA) refers to a method that affects the output of the model after some parameters in a mathematical model are changed, which can help researchers evaluate the influence of parameter estimation on uncertainly and provide a basis for further uncertainty analysis(such as probability analysis). This method is used to analyze the sensitivity of NDVI to various uncertain factors, find out the sensitivity factors and their maximum fluctuation range, and judge the most significant factors affecting NDVI. See lines 188-192.
(3) The image processing techniques (Lines 150-162 of the current version): The authors should not include any codes in their manuscript. Instead, they should convert the techniques into words or mathematical equations, which could be easily understood by layman. The authors should also explain the purpose of each step of their codes (Lines 150-162).
Re: Thanks for your suggestion. We have deleted the code in our manuscript, and add some information for the data process in lines 164-166.
- Figure 3 (Lines 263-266) is not professional, the authors should use consistent graphical software to obtain these types of graphs. Better follow the format of Figure 4. Further, the font size and style, labeling of y-axis of Figure 4 is inconsistent and problematic. Consistent font size, font style and positions of axes labeling should be used within the entire manuscript.
Re: The labeling of y-axis of Figure 4 have been revised consistent with other figures. And we consistent font size, font style and positions of axes labeling for all of figures except Figure 3 in manuscript, because we have made Figure 3 in the Matlab software, the word size and font in this software are different from those in excel(other figures).
(5) Figures 8(a) and 8(b) should be separated into 2 figures, because one is more on algorithmic explanation, and another on spatial plot.
Re: Done as suggestion.
(6) Lines 369-372: Additionally, compared with MODIS NDVI, Landsat NDVI has a longer time series and higher resolution, which can more accurately reflect the spatial distribution of surface vegetation and changes over the years, especially helpful for the monitoring of vegetation degradation areas - could the authors add in the exact spatial locations and geographical areas of these "vegetation degradation areas"?
Re: Thanks for your comment, we have added the geographical areas of vegetation degradation areas extracted by Landsat and MODIS, the new figure as following:
|
Satellite |
Landsat NDVI |
MODIS NDVI |
|
|
|
|
|
Legend |
|
|
Figure 12. The comparison diagram of significant change of Landsat NDVI, MODIS NDVI (2000-2018)
(7) Lines 422-424: The residual analysis and machine learning give us an opportunity to spatially analysis the impact of climate change and human activities on NDVI - How could this be conducted? It seems that the authors have not mentioned the potential extension in an explicit manner. They just outline the idea, but without providing much details. That's not a good practice for writing academic paper.
Re: The method such as residual analysis and CART model are all achieved pixel by pixel. That’s a advantage compared with the other statistical methods.
(8) Line 445: How do you define "important rate"? Any formula in the main text / Methodology?
Re: The “important rate” here refer to the contribution of each covariate in the random forest model, which implemented by randomForest::varlmp tools in R software. The usage of this tools as following:
varImp(object,...)
The object here is the RF model. The varlmp tool can help us to detect the contribution rate of the covariates to NDVI.
(9) The motivation of this study is not well explained in the "Introduction". Also, the Introduction should be re-written so that the intuition, motivation and ideas of development of this research could be more obviously illustrated. Further, is there any similar study conducted before? If so, please include them in the Introduction. Also, the structure of this manuscript should also be included in the last paragraph of Section I. Introduction. Please reformat the entire Introduction section.
Re: In the context of global change, vegetation growth can be easily affected by climate change and human activities. The study of vegetation dynamics and the response relationship between climate change and human activities and vegetation has become one of the key issue. Although research related to the spatio-temporal changes of vegetation coverage at global or regional scales has also yielded a series of phased result, the study like monitoring of vegetation coverage distribution and change by using medium-high resolution NDVI at large region still very few. At present, most study still applied the short time scale and low resolution dataset such as MODIS NDVI or GIMMS NDVI. We drived NDVI from the Landsat product with 30m resolution, that is the first intuition in our study.
Second intuition in our study is that the Wei and Jing River Basin have large difference in sediment content, we study the spatial distribution and change trend of NDVI in those two river basin, and help us to find the reasons of this difference.
(10) Section 4.3: The "consequences" of this research are not clearly explained. Do you actually mean "benefits" of this research in interdisciplinary areas of vegetation, land use distribution, etc.?
Re: This paper studies the temporal and spatial distribution changes of NDVI in the two river basins and their driving mechanisms. According to their temporal and spatial distribution and changes, we realize that the NDVI of the Jing and Wei rivers have a large difference, resulting in a large difference in sedimentation. However, the NDVI growth rate of the Jing River, which has a lower NDVI, is even higher than that of the Wei River, and the corresponding sediment content will also decrease. Will the sediment content of the two rivers tend to be the same in the future? Will the distinction between Jing and Wei disappear? Our research can lay a foundation for this, so this research is beneficial to the study of the relationship between vegetation and hydrology rather than "benefits" of this research in interdisciplinary areas of vegetation, land use distribution.
Finally, the authors should proceed for grammatical checking before re-submission.

Reviewer 3 Report
The authors removed the critical issues detected in the first and second step of review improving the level of their paper, which can now be published in the Environmental Research hand Public Health.
Author Response
We really appreciate the editors and reviewers for their comments and decisions and gave us an opportunity to submit my manuscript in this journal.